# BEV-VAE: A Unified BEV Representation for Generalizable Driving Scene Synthesis

## Abstract

Generative modeling has shown remarkable success in vision and language, inspiring research on synthesizing driving scenes. Existing multi-view synthesis approaches typically operate in image latent spaces with cross-attention to enforce spatial consistency, but they are tightly bound to camera configurations, which limits model generalization. We propose BEV-VAE, a variational autoencoder that learns a unified Bird's-Eye-View (BEV) representation from multi-view images, enabling encoding from arbitrary camera layouts and decoding to any desired viewpoint. Through multi-view image reconstruction and novel view synthesis, we show that BEV-VAE effectively fuses multi-view information and accurately models spatial structure. This capability allows it to generalize across camera configurations and facilitates scalable training on diverse datasets. Within the latent space of BEV-VAE, a Diffusion Transformer (DiT) generates BEV representations conditioned on 3D object layouts, enabling multi-view image synthesis with enhanced spatial consistency on nuScenes and achieving the first complete seven-view synthesis on AV2. Compared with training generative models in image latent spaces, BEV-VAE achieves superior computational efficiency. Finally, synthesized imagery significantly improves the perception performance of BEVFormer, highlighting the utility of generalizable scene synthesis for autonomous driving.

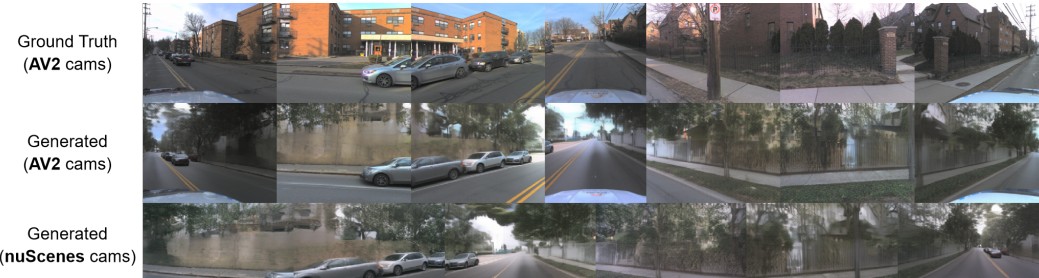

Figure 1: **Autonomous driving scene synthesis from AV2 to nuScenes.** BEV-VAE with DiT generates a BEV representation from 3D bounding boxes of AV2, which can then be decoded into multi-view images according to the camera configurations of nuScenes.

## 1 Introduction

The significant impact of generative modeling on vision (Rombach et al., 2022) and language (Achiam et al., 2023) has motivated research on the synthesis of driving scenes. Specifically, multi-view image synthesis conditioned on 3D object annotations can vary both object appearance and scene background while preserving the ground-truth 3D box locations. This enables 3D perception models (Li et al., 2024b) to learn the correspondence between changing visual appearance and fixed spatial positions. However, the effectiveness of such synthesized imagery critically depends on both per-view quality and cross-view consistency. Existing approaches (Li et al., 2024a; Gao et al., 2023; Wen et al., 2024; Wang et al., 2024) typically achieve multi-view synthesis by training generative models in the image latent space, ensuring spatial consistency through cross-view attention. Although this paradigm can ensure consistency, it introduces significant computational costs and high modeling complexity. Moreover, it is inherently tied to specific vehicle types and camera layouts, limiting both the scale of available training data and the generalizability of the synthesized imagery. For example, a model trained on seven camera views cannot be applied directly to vehicles equipped with six.

Table 1: **Comparison of autonomous driving datasets with full 360° multi-camera coverage.** These datasets vary in dataset scale, camera configurations, 3D annotation categories, and recording locations, where WS101 does not provide 3D annotations.

| Dataset | # Frames | # Cameras | # Classes | Recording Locations |
|---------|----------|-----------|-----------|---------------------|
| WS101 | 17K | 5 | 0 | London, San Francisco Bay Area |
| nuScenes | 155K | 6 | 23 | Boston, Pittsburgh, Las Vegas, Singapore |
| AV2 | 224K | 7 | 30 | Austin, Detroit, Miami, Pittsburgh, Palo Alto, Washington DC |
| nuPlan | 3.11M | 8 | 7 | Boston, Pittsburgh, Las Vegas, Singapore |

In reality, multi-view images with varying camera layouts are only different projections of the same scene. Motivated by this insight, we introduce BEV-VAE, a variational autoencoder that learns a unified BEV representation from multi-view images and utilizes this latent space for generative modeling. The BEV representation integrates the semantics of all views and constructs the 3D structure of the scene, enabling encoding from arbitrary camera layouts and decoding to any desired viewpoints. It avoids explicitly modeling spatial relationships across views, which substantially reduces computational cost and modeling complexity for generative modeling. In addition, training can be performed on multiple datasets that cover different types of vehicle and camera layouts. This overcomes the data isolation limitations of existing methods and enables generalizable driving scene synthesis across datasets and viewpoints.

We systematically evaluate the generalizability of BEV-VAE across four autonomous driving datasets (Zürn et al., 2024; Caesar et al., 2020; Wilson et al., 2023; Caesar et al., 2021), which vary in dataset scale, camera configurations and recording locations (see Tab. 1). The spatial modeling capability of BEV-VAE is validated by multi-view image reconstruction, as the reconstruction fidelity reflects its ability to construct the spatial relationships between objects and the background in the scene. Novel view synthesis is further achieved by modifying camera poses when decoding the BEV representation into images, directly demonstrating that BEV-VAE encodes precise spatial structure and comprehensive scene semantics. In addition, BEV-VAE overcomes the data isolation caused by varying vehicle camera setups, effectively integrating datasets collected worldwide and greatly increasing the diversity of training data. Models trained on mixed datasets achieve significantly higher reconstruction quality than trained individually on AV2 or nuScenes, demonstrating the scalability of BEV-VAE. Meanwhile, multi-dataset joint training enables BEV-VAE to generalize across different vehicle types and camera setups. For example, it can convert images from the 8-camera configuration of nuPlan to the 7-camera setup of AV2 or the 6-camera setup of nuScenes. This indicates that BEV-VAE generalizes not only across camera poses but also camera intrinsics. Furthermore, BEV-VAE enables few-shot adaptation on WS101 by leveraging pretraining on diverse camera configurations, and achieves significantly improved reconstruction quality after fine-tuning.

We train a Diffusion Transformer (DiT) (Peebles & Xie, 2023) in the latent space of BEV-VAE to enable multi-view image synthesis conditioned on 3D object layouts. These object layouts are encoded as occupancy grids that are spatially aligned with the BEV representation, allowing precise specification of object positions and counts in the scene, analogous to ControlNet (Zhang et al., 2023). Specifically, we achieve multi-view image synthesis with enhanced spatial consistency on nuScenes, and are the first to synthesize images for all seven camera views on AV2. Furthermore, the unified BEV representation enables direct cross-dataset viewpoint conversion by decoding AV2-synthesized scenes with the camera configuration of nuScenes. By operating in this compact BEV latent space, rather than maintaining a collection of image representations, our method substantially reduces GPU memory consumption and inference latency. Finally, we show that synthesized imagery can significantly improve the performance of BEVFormer on nuScenes, validating the effectiveness of synthesis-based appearance diversification as a data augmentation strategy for perception.

## 2 RELATED WORK

### 2.1 BIRD'S-EYE-VIEW REPRESENTATION

Autonomous driving relies on Bird's Eye View (BEV) to integrate information from multiple camera perspectives. The construction of BEV representations is typically approached in two ways: bottom-up and top-down. Bottom-up methods (Philion & Fidler, 2020; Huang et al., 2021; Liu et al., 2023) estimate depth to lift 2D features into 3D space before fusing them into BEV. In contrast, top-down methods (Li et al., 2024b; Hu et al., 2023) employ deformable attention (DA) and query mechanisms to efficiently aggregate features through dynamic sampling of key regions. These methods learn BEV

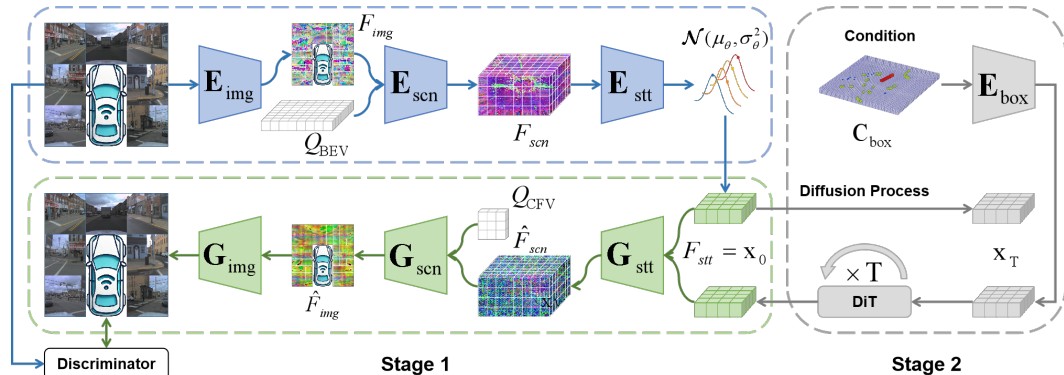

Figure 2: **Overall architecture of BEV-VAE with DiT for autonomous driving scene synthesis.**
In Stage 1, BEV-VAE learns to encode multi-view images into a compact latent space in BEV and
reconstruct them, modeling the spatial structure and representing the scene semantics. In Stage 2, DiT
is trained with Classifier-Free Guidance (CFG) in this latent space to generate BEV representations
from random noise, which are then decoded into multi-view images.

representations from perception tasks. BEVWorld (Zhang et al., 2024) leverages NeRF (Mildenhall
et al., 2021) rendering, while SelfOcc (Huang et al., 2024) models SDF fields with volumetric
rendering, to learn BEV representations via image reconstruction.

## 2.2 VARIATIONAL AUTOENCODER FOR GENERATIVE MODELING

VAE provides an efficient latent-variable framework for generative modeling. VQ-VAE (Van
Den Oord et al., 2017) introduces discrete codebooks, enabling Transformer-based autoregressive
image generation. VQGAN (Esser et al., 2021) enhances visual fidelity through adversarial and
perceptual losses (Johnson et al., 2016), while ViT-VQGAN (Yu et al., 2021) adopts ViT (Dosovit-
skiy, 2020) architectures to improve global context modeling and codebook utilization. In parallel,
diffusion models (Ho et al., 2020; Song et al., 2020; Rombach et al., 2022) achieve high-quality
synthesis via iterative denoising, and DiT (Peebles & Xie, 2023) unifies diffusion and Transformer
architectures for scalable generative modeling. Despite these advances, improved VAE variants
remain critical, offering higher compression (Chen et al., 2024) and better alignment with foundation
model representations (Yao et al., 2025), supporting scalable and effective generative modeling.

## 2.3 AUTONOMOUS DRIVING SCENE SYNTHESIS

Autonomous driving scene synthesis is predominantly formulated as a multi-view generation problem,
where 3D scenes are implicitly represented by multiple 2D images. BEVGen (Swerdlow et al.,
2024) employs autoregressive modeling to generate multi-view images conditioned on BEV lay-
outs, injecting camera direction vectors and BEV features as an attention bias to improve spatial
consistency. Recent advances shift toward diffusion-based frameworks by adapting Stable Diffusion
for autonomous driving. Methods such as DrivingDiffusion (Li et al., 2024a), MagicDrive (Gao
et al., 2023), and Panacea (Wen et al., 2024) utilize cross-attention on adjacent view images to
ensure consistency between perspectives. MagicDrive integrates camera pose information by en-
coding camera parameters similar to NeRF (Mildenhall et al., 2021), while Panacea extends this
approach by generating pseudo-RGB images of camera frustum directions and embedding pose
information through ControlNet (Zhang et al., 2023). Additionally, DriveWM (Wang et al., 2024)
uses self-attention to fuse spatially aligned features across views and predicts stitched views between
nonadjacent references to maintain multi-view spatial consistency. Despite these advances, existing
methods largely underexploit explicit camera geometry and lack structured 3D scene modeling,
confining generation to fixed viewpoints. This limits viewpoint flexibility and hampers cross-platform
generalization in autonomous driving scenarios.

# 3 METHOD

## 3.1 OVERALL ARCHITECTURE OF BEV-VAE

BEV-VAE consists of a Transformer-based encoder $E$, decoder $G$, and a StyleGAN discriminator $D$. The encoder $E$ maps multi-view images into a latent Gaussian distribution via its image, scene, and state encoders, from which state features are sampled via reparameterization. The decoder $G$, comprising state, scene, and image decoders, reconstructs spatially consistent multi-view images from the state features. The discriminator $D$ distinguishes real from reconstructed images, guiding $G$ with adversarial loss. Both encoder $E$ and decoder $G$ are trained with KL divergence, reconstruction, and adversarial losses.

### 3.1.1 ENCODER

**Image Encoder** employs ViT with a patch size of 8 to encode a $256 \times 256$ image into a $32 \times 32$ token sequence. To capture semantic information and local details for 3D scene encoding, an upsampling-only FPN Lin et al. (2017) constructs a three-level feature pyramid to enhance multi-scale representation. The process can be formulated as: $F_{img} = \text{FPN}(\mathbf{E}_{img}(x)) = \text{Concat}(F^0_{img}, F^1_{img}, F^2_{img})$, where $F^i_{img} \in \mathbb{R}^{V \times L_i \times C}(i \in [0, 2])$ are the multi-scale flattened image features with $C = 96$ and sequence length $L_i = 32 \times 32 \times 2^{2i}$. Here, $V$ is the number of views.

**Scene Encoder** utilizes a deformable attention mechanism to construct 3D scene features by extracting multiview image features. A $128 \times 128$ grid of pillars is pre-defined around the ego vehicle in BEV, each with a height of 8. All reference points in the same pillar share a learnable query, while different height positions are distinguished through positional encoding. The reference points of scene features are projected onto image features by camera parameters, enabling BEV queries to aggregate spatially aligned features from multiview image features via deformable attention. The process can be formulated as: $F_{scn} = \frac{1}{|\mathcal{V}_{\text{hit}}|} \sum_{v \in \mathcal{V}_{\text{hit}}} \text{DA}(Q_{\text{BEV}}, P_{\text{BEV}}, F^{(v)}_{img})$, where $Q_{\text{BEV}} \in \mathbb{R}^{L_Q \times C}$ are the flattened 3D BEV queries with $C = 96$, $P_{\text{BEV}} \in \mathbb{R}^{L_Q \times 3}$ denote the corresponding reference points, $F^{(v)}_{img} \in \mathbb{R}^{L_V \times C}$ is the image feature sequence of the view $v$, and the set $\mathcal{V}_{\text{hit}}$ refers to the views containing projected reference points, ensuring that only relevant views contribute to the aggregated scene feature. Here, $L_Q = 8 \times 128 \times 128$ is the BEV query sequence length, and $L_V = \sum_{i=0}^{2}(32 \times 32 \times 2^{2i})$ is the total image feature sequence length across resolutions.

**State Encoder** integrates multi-height scene features in BEV by concatenating them along the height dimension, reshaping the input from $96 \times 8 \times 128 \times 128$ to $768 \times 128 \times 128$. It then partitions the features into $32 \times 32$ patches along the horizontal plane, reducing the computational cost while introducing local receptive fields. Finally, it applies self-attention to model global spatial relationships and encode highly compressed spatial state features.

### 3.1.2 DECODER

**State Decoder** is responsible for reconstructing structurally detailed 3D scene features from the compressed 2D state representation $F_{stt}$, which is the BEV representation obtained after VAE reparameterization and is the actual input used to train DiT. It first applies self-attention to capture global spatial relationships, and then regroups the features to restore horizontal and height structures. The state features are first expanded from $32 \times 32$ to $128 \times 128$ along the horizontal plane through deconvolution, then further transformed from $768 \times 128 \times 128$ to the original multi-height format $96 \times 8 \times 128 \times 128$ through dimension partitioning. To refine 3D scene feature decoding, a downsampling-only FPN is employed, effectively reconstructing detailed structures across scales. The process can be formulated as: $\hat{F}_{scn} = \text{FPN}(\mathbf{G}_{stt}(\hat{x})) = \text{Concat}(\hat{F}^0_{scn}, \hat{F}^1_{scn}, \hat{F}^2_{scn})$, where $\hat{F}^i_{scn} \in \mathbb{R}^{L_i \times C}(i \in [0, 2])$ are the reconstructed multi-scale flattened scene features with $C = 96$ and sequence length $L_i = 8 \times 128 \times 128 \times 2^{-3i}$.

**Scene Decoder** transforms scene features from the Bird's Eye View (BEV) to the Camera's Frustum View (CFV) and aggregates multi-depth information to reconstruct image features. A $32 \times 32$ frustum of rays is predefined per camera, each spanning 60 depth levels. All reference points along the same ray share a learnable query, while different depth positions are distinguished through positional encoding. Similar to the projection of reference points of scene features from BEV

onto image features via camera parameters, reference points of scene features in CFV can also be projected to BEV, enabling CFV queries to construct features along depth dimensions for different views via deformable attention. Furthermore, CFV queries estimate depth weights to perform a weighted summation of the features at all reference points along the ray, thereby generating the projected image features. Considering that some reference points may exceed the range of scene features, their corresponding weights are set to 0. The process can be formulated as: $\hat{F}_{img}^{(v)} = \sum_{d \in \mathcal{D}_{hit}} W_d \odot \mathrm{DA}(Q_{\mathrm{CFV}}, P_{\mathrm{CFV}}, \hat{F}_{scn})$, where $Q_{\mathrm{CFV}} \in \mathbb{R}^{L_Q \times C}$ are the flattened 3D CFV queries with $C = 96$, $P_{\mathrm{CFV}} \in \mathbb{R}^{L_Q \times 3}$ denote the corresponding reference points, $\hat{F}_{scn} \in \mathbb{R}^{L_V \times C}$ is the reconstructed scene feature sequence, and the set $\mathcal{D}_{hit}$ refers to the depth positions along the ray where reference points fall within the valid scene feature range, ensuring that only effective depth positions contribute to the aggregated image feature. Here, $L_Q = 60 \times 32 \times 32$ is the CFV query sequence length, and $L_V = \sum_{i=0}^{2}(8 \times 128 \times 128 \times 2^{-3i})$ is the total reconstructed scene feature sequence length across resolutions.

**Image Decoder** progressively restores pixel-level details by processing scene features projected onto the image plane. As its preceding stage, the scene decoder aggregates scene features along the ray depth dimension but lacks interactions between rays. To complement this, it maps the projected scene features ($C = 96$) to 768 dimensions via a linear layer, models global spatial and semantic relationships on the image plane by self-attention, and upscales the resolution from $32 \times 32$ to $256 \times 256$ with deconvolution, reconstructing fine-grained image details.

### 3.1.3 Loss

**KL Divergence Loss** regularizes the latent distribution of the state features, enforcing closeness to a standard normal distribution and ensuring continuity in the latent space: $\mathcal{L}_{\mathrm{KL}} = D_{\mathrm{KL}}(q_\phi(z \mid x)\|p(z)) = \frac{1}{2}\sum_{i=1}^{d}(\sigma_i^2 + \mu_i^2 - 1 - \log \sigma_i^2)$, where $p(z)$ is defined as $\mathcal{N}(0, I)$, $d$ is the dimension of state features, and $\mu_i, \sigma_i^2$ are the mean and variance of the $i$-th latent dimension predicted by the encoder $E$. To allow gradient-based optimization of the stochastic sampling process, the reparameterization trick is used. Instead of directly sampling $z$ from $q_\phi(z \mid x)$, it is reparameterized as: $z = \mu + \sigma \odot \epsilon, \quad (\mu, \sigma) = E(x), \quad \epsilon \sim \mathcal{N}(0, I)$.

**Reconstruction Loss** ensures that the reconstructed image $\hat{x} = G(z)$ retains both pixel-level details and high-level semantic structure of the target image $x$. This is achieved by combining pixel-wise loss with perceptual loss: $\mathcal{L}_{\mathrm{R}} = \mathcal{L}_2 + \mathcal{L}_{\mathrm{perceptual}} = \|x - \hat{x}\|^2 + \sum_l \|\psi_l(x) - \psi_l(\hat{x})\|^2$. Here, $\mathcal{L}_2$ enforces pixel-wise similarity between the image $x$ and its reconstruction $\hat{x}$, while $\mathcal{L}_{\mathrm{perceptual}}$ captures structural and semantic consistency by comparing feature maps $\psi_l(x)$ and $\psi_l(\hat{x})$ extracted from the $l$-th layer of a pre-trained VGG-16. This balance preserves fine details and perceptual coherence, yielding realistic reconstructions.

**Discriminator Loss** enables the discriminator $D$ to distinguish real images from reconstructed ones, improving its ability to provide meaningful adversarial feedback. With the hinge loss formulation, it is expressed as: $\mathcal{L}_{\mathrm{D}} = \max(0, 1 - D(x)) + \max(0, 1 + D(\hat{x}))$, which encourages the discriminator to assign higher scores to real images and lower scores to reconstructed ones. Hinge loss stabilizes adversarial training by preventing excessively large gradients for confident predictions while ensuring effective feedback for refining reconstruction quality, leading to more stable and efficient optimization.

**Adversarial Loss** leverages the discriminator's feedback to enhance the perceptual realism of reconstructed images and is defined as: $\mathcal{L}_{\mathrm{A}} = -D(\hat{x})$

**Total Loss for Encoder and Decoder** combines the KL divergence loss, reconstruction loss, and adversarial loss, ensuring effective latent space regularization and perceptual realism. It is formulated as: $\mathcal{L}_{\mathrm{G}} = \beta \cdot \mathcal{L}_{\mathrm{KL}} + \mathcal{L}_{\mathrm{R}} + 0.1 \cdot \lambda \cdot \mathcal{L}_{\mathrm{A}}$ where $\beta = 10^{-6}$ controls the strength of the KL divergence regularization. The adaptive weight $\lambda$ balances the adversarial loss relative to the reconstruction loss, ensuring that the adversarial term contributes meaningfully without overpowering reconstruction. It is computed as $\lambda = \frac{\nabla_{G_L}[\mathcal{L}_{\mathrm{R}}]}{\nabla_{G_L}[\mathcal{L}_{\mathrm{A}}]+\delta}$ with $\nabla_{G_L}[\cdot]$ denoting the gradient of the corresponding term with respect to the last layer $L$ of the decoder, and $\delta = 10^{-6}$ ensuring numerical stability.

Figure 3: **Multi-view image reconstruction on nuPlan.** Row 1 shows real images from the nuPlan validation set and Row 2 shows the corresponding reconstructions. Pedestrians, traffic lights, trucks, trailers, cars, crosswalks, and road markings are faithfully reconstructed.

Table 2: **BEV-VAE vs. SD-VAE in multi-view reconstruction.** SD-VAE focuses on per-view image fidelity, whereas PAS-trained BEV-VAE achieves superior multi-view spatial consistency (MVSC).

(a) Reconstruction metrics on nuScenes compared with SD-VAE.

| Model | Training | Validation | PSNR↑ | SSIM↑ | MVSC↑ | rFID↓ |
|---|---|---|---|---|---|---|
| SD-VAE | LAION-5B | nuScenes | **29.63** | **0.8283** | 0.9292 | **2.18** |
| BEV-VAE | nuScenes | nuScenes | 26.13 | 0.7231 | 0.9250 | 6.66 |
| BEV-VAE | PAS | nuScenes | 28.88 | 0.8028 | **0.9756** | 4.74 |

(b) Reconstruction metrics on AV2 compared with SD-VAE.

| Model | Training | Validation | PSNR↑ | SSIM↑ | MVSC↑ | rFID↓ |
|---|---|---|---|---|---|---|
| SD-VAE | LAION-5B | AV2 | **27.81** | **0.8229** | 0.8962 | **1.87** |
| **BEV-VAE** | AV2 | AV2 | 26.02 | 0.7651 | 0.9197 | 4.15 |
| **BEV-VAE** | PAS | AV2 | 27.29 | 0.8028 | **0.9461** | 2.82 |

## 3.2 SPATIALLY-ALIGNED BEV GENERATION FROM 3D OBJECT LAYOUTS

**BEV-VAE w/ DiT** extends BEV-VAE by integrating DiT in its latent space, leveraging CFG to enhance conditional generation. By explicitly incorporating structured occupancy constraints from 3D object bounding boxes, it ensures spatial consistency and controllability in generation. Given a set of 3D bounding boxes $\{\mathbf{b}_i\}_{i=1}^{N}$, each parameterized as: $\mathbf{b} = (q_w, q_x, q_y, q_z, x_c, y_c, z_c, l, w, h, c)$, where the quaternion $q = (q_w, q_x, q_y, q_z)$ encodes the 3D orientation, $(x_c, y_c, z_c)$ specifies the box center in the ego coordinate system, $(l, w, h)$ represents the size of the box, and $c \in 1, \ldots, C$ is the semantic class index. These boxes are voxelized into a binary occupancy tensor $\mathbf{C}_{\text{box}} \in \{0,1\}^{C \times 8 \times 128 \times 128}$, where each voxel represents whether a given spatial location is occupied by a bounding box of a particular class. Formally, it is defined as: $\mathbf{C}_{\text{box}}(c, z, y, x) = \max_{i:c_i=c} \mathbf{1}[(z, y, x) \in \Omega(\mathbf{b}_i)]$ where $\mathbf{1}[\cdot]$ is an indicator function, and $\Omega(\mathbf{b}_i)$ denotes the discretized voxelized representation of bounding box $\mathbf{b}_i$. The max operation aggregates occupancy information from overlapping bounding boxes within the same class. The occupancy tensor $\mathbf{C}_{\text{box}}$ is downsampled via non-overlapping patch partitioning in the BEV plane, yielding a feature of shape $96 \times 8 \times 32 \times 32$, followed by channel-wise concatenation of the height dimension to form the conditional occupancy feature $F_{box} \in \mathbb{R}^{768 \times 32 \times 32}$. Aligned with the state feature $F_{\text{stt}}$, it is injected via element-wise addition: $F'_{stt} = F_{stt} + s \cdot F_{box}$, where $s$ is the guidance scale in CFG. This ensures spatial consistency by aligning the conditional occupancy features and state features within the shared BEV coordinate system, allowing DiT to focus on relevant regions by explicitly incorporating object category and location information.

## 4 EXPERIMENTS

Table 3: **Few-shot reconstruction metrics on WS101 compared with SD-VAE.**

| Model | Training | Validation | PSNR↑ | SSIM↑ | MVSC↑ | rFID↓ |
|---|---|---|---|---|---|---|
| SD-VAE | LAION-5B | WS101 | 23.38 | **0.7050** | 0.8580 | **4.59** |
| BEV-VAE | PAS | WS101 | 16.6 | 0.3998 | 0.8309 | 56.7 |
| BEV-VAE | PAS+WS101 | WS101 | **23.46** | 0.6844 | **0.9505** | 13.78 |

### 4.1 DATASETS

This study uses four multi-camera autonomous driving datasets that differ substantially in scale, camera configuration, annotated categories, and recording locations, as shown in Tab. 1. Despite these differences, all datasets provide full 360° coverage of the surrounding scene.

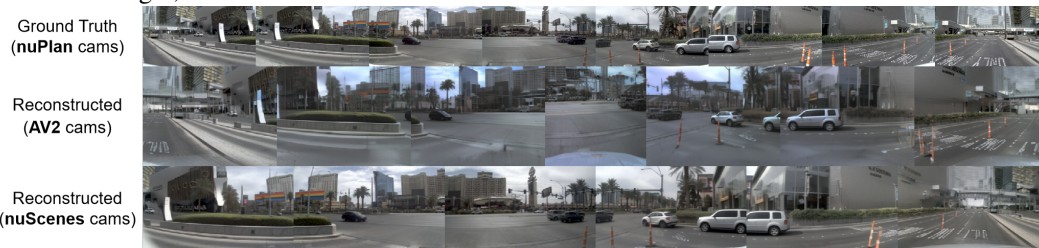

Figure 4: **Novel view synthesis via camera pose modifications on nuScenes.** Row 1 shows real images from the nuScenes validation set, and Rows 2-3 show reconstructions with all cameras rotated 30° left and right, where the cement truck and tower crane truck remain consistent across views.

Figure 5: **Novel view synthesis cross camera configurations.** Row 1 presents real images from the nuPlan validation set. Row 2 and Row 3 show reconstructions using camera parameters from AV2 and nuScenes, respectively. The model captures dataset-specific vehicle priors: AV2 include both the front and rear of the ego vehicle, while nuScenes mainly show the rear.

**The WS101 dataset** (Zürn et al., 2024) consists of 5 cameras with 101 scenes. We use the first 84 scenes as the training set and the remaining 17 scenes as the validation set. Each scene contains approximately 200 samples. Note that 3D object bounding boxes are not provided.

**The nuScenes dataset** (Caesar et al., 2020) consists of 6 cameras with 700 training scenes and 150 validation scenes. Each scene contains approximately 220 samples, of which 40 are annotated across 10 object categories. In total, it includes 155k training samples, of which 28k are annotated, and 33k validation samples, of which 6k are annotated.

**The AV2 dataset** (Wilson et al., 2023) consists of 7 cameras, with the front camera rotated by 90°. It includes 700 training scenes and 150 validation scenes. Each scene contains approximately 300 samples, of which 150 are annotated across 30 object categories. In total, it includes 224k training samples, of which 109k are annotated, and 47k validation samples, of which 23k are annotated.

**The nuPlan dataset** (Caesar et al., 2021) consists of 8 cameras with 1085 training logs. The training set comprises 3.11 million samples annotated with 7 object categories, but we only use the images from the training set.

## 4.2 SETTINGS

We introduce a new hybrid autonomous driving dataset configuration, **PAS**, which combines nu**P**lan, **A**V2, and nu**S**cenes. The training process consists of two stages, all using the AdamW optimizer with a learning rate of 1e-4 and a 5k-step warm-up.

**Stage 1:** Training is performed on **PAS** with a batch size of 1 per GPU for 800k iterations on 8 NVIDIA H100 GPUs. The optimization settings are $\beta = (0.9, 0.99)$, weight decay $1e{-}4$, and EMA decay 0.9999.

**Stage 2:** Training is conducted on **PAS** with a batch size of 8 per GPU for 200k iterations with 3D object annotations from AV2 or 400k iterations with annotations from nuScenes, using 8 NVIDIA A800 GPUs. The optimization settings are $\beta = (0.9, 0.95)$, weight decay 0.1, and EMA decay 0.999.

## 4.3 METRICS

The performance of BEV-VAE is evaluated using multiple metrics covering reconstruction quality, multi-view spatial consistency, and generation quality.

Table 4: **Comparison of multi-view image generation on nuScenes.**

| Metric | BEVGen | Panacea | MagicDrive | DrivingDiffusion | DriveWM | Ours |
|--------|--------|---------|------------|------------------|---------|------|
| gFID↓ | 25.54 | 16.96 | 16.20 | 15.83 | **12.99** | 20.7 |
| MVSC↑ | - | 0.9189 | 0.8310 | - | - | **0.9310** |

Table 5: **Impact of CFG scale on gFID for multi-view image generation.**

| Dataset | 0 | 1 | 2 | 3 | 4 | 5 | 6 | 7 | 8 | 9 |
|---------|---|---|---|---|---|---|---|---|---|---|
| nuScenes | 48.99 | 35.61 | 23.34 | 20.73 | **20.70** | 21.03 | 21,38 | 21.74 | 21.91 | 22.18 |
| AV2 | 56.28 | 24.84 | 18.96 | 17.02 | 16.06 | 15.77 | **15.73** | 16.19 | 16.61 | 17.16 |

**PSNR** and **SSIM** measure the similarity between reconstructed and original images, with PSNR assessing signal fidelity and SSIM focusing on structural consistency.

**Multi-View Spatial Consistency (MVSC)** evaluates spatial consistency in multi-view reconstruction. Following BEVGen Swerdlow et al. (2024) and DriveWM Wang et al. (2024), a pre-trained LoFTR Sun et al. (2021) is used to compute keypoint matching confidence between adjacent views. MVSC is the ratio of average adjacent-view matching confidence in reconstructed images to that in real images, where higher values imply better alignment.

**FID** and **FVD** are used to evaluate the quality of generated data in a deep feature space. FID measures the fidelity of reconstructed and generated multi-view images, while FVD assesses the temporal consistency and realism of generated front-view video sequences.

### 4.4 MULTI-VIEW IMAGE RECONSTRUCTION

BEV-VAE learns unified BEV representations by reconstructing multi-view images, integrating semantics from all camera views while modeling 3D spatial structure. Reconstruction metrics provide an indirect evaluation of the quality of the learned BEV representations. For reference, we compare with SD-VAE (Rombach et al., 2022), a foundational model trained on LAION-5B Schuhmann et al. (2022), which encodes a single $256 \times 256$ image into a $32 \times 32 \times 4$ latent. In contrast, BEV-VAE encodes multiple $256 \times 256$ views into a $32 \times 32 \times 16$ BEV latent, facing the more challenging task of modeling underlying 3D structure. As shown in Tab. 2, BEV-VAE trained on nuScenes or AV2 alone underperforms SD-VAE. However, when trained on the hybrid PAS dataset that combines multiple autonomous driving datasets with diverse camera configurations, BEV-VAE achieves a notable improvement, surpassing SD-VAE by a large margin on MVSC. This demonstrates that BEV-VAE effectively integrates multi-view semantics and captures spatial structure. Moreover, as illustrated in Fig. 3, BEV-VAE reconstructs most elements of complex driving scenes with high fidelity, while decoupling per-view reconstruction quality from cross-view spatial consistency: since all views are decoded from the same BEV representation, spatial consistency across views is guaranteed regardless of per-view reconstruction quality.

### 4.5 NOVEL VIEW SYNTHESIS

Reconstruction metrics provide a quantitative proxy for evaluating the quality of BEV representations, but they cannot directly verify whether BEV-VAE accurately models the spatial structure of objects and background from multi-view semantics. Conversely, if BEV-VAE captures such spatial structures correctly, it should be able to synthesize novel views simply by adjusting the camera poses, as illustrated in Fig. 4. Furthermore, leveraging the hybrid PAS dataset configuration, BEV-VAE demonstrates generalization not only to unseen camera poses but also to varying camera intrinsics, enabling the reconstruction of nuPlan scenes under the camera configurations of AV2 or nuScenes, as shown in Fig. 5.

### 4.6 FEW-SHOT ADAPTATION

BEV-VAE learns unified BEV representations that generalizes across diverse camera configurations. When applied to previously unseen camera setups, the pretrained model provides a strong initialization for reconstruction. Leveraging the learned spatial priors, BEV-VAE can be efficiently adapted to new domains with limited data. Fine-tuning on WS101 for 50k iterations under the new camera

configuration significantly improves reconstruction quality, outperforming SD-VAE in both PSNR and MVSC, as shown in Tab. 3.

## 4.7 AUTONOMOUS DRIVING SCENE SYNTHESIS

As shown in Fig. 1, BEV-VAE w/ DiT generates BEV representations from 3D object layouts that can be decoded to arbitrary viewpoints, enabling a single model to support vehicles with different camera setups and achieve cross-platform scene generalization. We compare our approach with prior multi-view image generation methods in Tab. 4. Although our method has a higher gFID than previous works, it demonstrates superior multi-view spatial consistency. CFG scale ablation (Tab. 5) shows that the optimal gFID is achieved at a scale of 4 for nuScenes (20.7) and 6 for AV2 (15.73).

## 4.8 COMPUTATIONAL EFFICIENCY

As shown in Tab. 6, the GPU memory usage of BEV-VAE w/ DiT and MagicDrive is benchmarked on an A800 across different batch sizes. At batch size 4, MagicDrive nearly exhausts the 80 GB memory capacity, whereas BEV-VAE w/ DiT scales up to batch size 32. As shown in Tab. 7, inference latency is further evaluated on an RTX 3090 using 20-step DDIM sampling, where BEV-VAE w/ DiT achieves a 4× speedup over MagicDrive, even without enabling BF16 or Flash Attention.

Table 6: **GPU memory usage (GB) on A800 for nuScenes across different batch sizes.**

| Model | 1 | 2 | 4 | 8 | 16 | 32 |
|---|---|---|---|---|---|---|
| MagicDrive | 26.4 | 42.1 | 73.5 | OOM | OOM | OOM |
| BEV-VAE w/ DiT | 9.5 | 11.8 | 16.2 | 25.2 | 43.1 | 79.2 |

Table 7: **Inference latency (s) on RTX 3090 for nuScenes.**

| Model | Base | Flash Attention | BF16 | Flash Attention + BF16 |
|---|---|---|---|---|
| MagicDrive | 5.381 | - | - | - |
| BEV-VAE w/ DiT | 1.160 | 1.123 | 0.930 | 0.872 |

## 4.9 DATA AUGMENTATION FOR PERCEPTION

BEV-VAE w/ DiT using the Historical Frame Replacement strategy (randomly replacing real frames with generated ones) improves BEVFormer's perception by enabling the model to learn invariance of object locations relative to appearance. Compared to BEVGen, which augments the dataset by adding synthetic data, our approach requires no additional computational cost while achieving the highest NDS, as shown in Tab. 8.

Table 8: **Perception performance with generative augmentation.**

| Perception Model | Generative Model | Augmentation Strategy | mAP↑ | NDS↑ |
|---|---|---|---|---|
| BEVFormer Tiny | - | - | 25.2 | 35.4 |
| BEVFormer Tiny | BEVGen | Training Set + 6k Synthetic Data | **27.3** | 37.2 |
| BEVFormer Tiny | BEV-VAE w/ DiT | Historical Frame Replacement | 27.1 | **37.4** |

## 5 CONCLUSION

In this paper, we present BEV-VAE, a variational autoencoder that learns a unified BEV representation from multi-view images, capturing both scene semantics and 3D structure. BEV-VAE supports encoding from arbitrary camera layouts and decoding to any desired viewpoints, enabling scalable training across datasets with different camera configurations. Within the latent space of BEV-VAE, DiT can generate BEV representations conditioned on 3D object layouts, which can also be decoded to arbitrary viewpoints, allowing cross-platform generalizable applications. Moreover, this synthesized imagery significantly enhances the performance of downstream perception models. Although BEV-VAE does not surpass previous methods in FID for multi-view image synthesis, this is partly due to the greater difficulty of generating full scenes compared with fixed-view images. In the future, we plan to extend BEV-VAE to temporal scenarios.

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

# Supplementary Material for BEV-VAE

The supplementary material provides additional context and experimental results that complement the main paper on BEV-VAE. Sec. A introduces the fundamental principles of the generative models used in our framework, while Sec. B explains the multi-view spatial consistency (MVSC) metric in detail. Sec. C presents visualizations of multi-view image reconstruction under few-shot adaptation on the WS101 dataset. Sec. D showcases fine-grained control of 3D object layouts, enabling flexible adjustment of the number and spatial positions of vehicles. Sec. E further provides quantitative analyses of novel view synthesis, and Sec. F reports a detailed breakdown of per-module inference latency in BEV-VAE. Finally, Sec. G discusses the challenges and potential directions for extending the framework to 512×512 image resolution.

## A   PRELIMINARY FOR GENERATIVE MODELS

**VAE** is trained by maximizing the Evidence Lower Bound (ELBO) as follows:

$$\log p_\theta(x) \geq \mathbb{E}_{q_\phi(z|x)}\left[\log p_\theta(x|z)\right] - D_{\mathrm{KL}}\left(q_\phi(z|x) \,\|\, p_\theta(z)\right), \tag{1}$$

where $x$ is the input data, $z$ is the latent variable, $\phi$ and $\theta$ are the encoder and decoder parameters, respectively. The first term ensures that the decoder $p_\theta(x \mid z)$ can accurately reconstruct $x$ from the latent variable $z$, and the second term penalizes the divergence between the posterior $q_\phi(z \mid x)$ and the prior $p(z)$, typically $\mathcal{N}(0, I)$, encouraging a structured and continuous latent space.

**Diffusion models** define a forward process that gradually adds Gaussian noise to real data $x_0$, formulated as:

$$q(x_t \mid x_0) = \mathcal{N}(x_t; \sqrt{\bar{\alpha}_t}x_0, (1 - \bar{\alpha}_t)\mathbf{I}), \tag{2}$$

where $\bar{\alpha}_t$ are pre-defined noise scheduling coefficients, enabling direct sampling of $x_t$ from $x_0$ without iterative noise application. With reparameterization, the noised sample is:

$$x_t = \sqrt{\bar{\alpha}_t}x_0 + \sqrt{1 - \bar{\alpha}_t}\epsilon_t, \quad \epsilon_t \sim \mathcal{N}(0, \mathbf{I}). \tag{3}$$

This highlights the relationship between $x_0$ and noise $\epsilon_t$, enabling training via noise prediction. The reverse process learns to iteratively denoise $x_t$ back to $x_0$, where

$$p_\theta(x_{t-1} \mid x_t) = \mathcal{N}(x_{t-1}; \mu_\theta(x_t), \sigma_t^2\mathbf{I}), \tag{4}$$

The mean $\mu_\theta(x_t)$ is predicted by the model, while the variance $\sigma_t^2$ is fixed as in DDPM. The ELBO is minimized during training, simplifying to a noise prediction objective:

$$\mathcal{L}_{\mathrm{simple}}(\theta) = \mathbb{E}[\|\epsilon_\theta(x_t) - \epsilon_t\|_2^2]. \tag{5}$$

Sampling starts from a standard Gaussian $x_T \sim \mathcal{N}(0, \mathbf{I})$ and iteratively denoises via $p_\theta(x_{t-1} \mid x_t)$ to generate samples consistent with the target distribution.

**Classifier-Free Guidance (CFG)** enhances conditional diffusion models by adjusting the sampling process to prioritize samples with high $p(c \mid x)$. By applying Bayes' rule, the gradient formulation is derived as:

$$\nabla_x \log p(c \mid x) = \nabla_x \log p(x \mid c) - \nabla_x \log p(x), \tag{6}$$

which implies that increasing $p(c \mid x)$ can be achieved by adjusting the diffusion trajectory toward higher $p(x \mid c)$. The reverse diffusion process follows:

$$p_\theta(x_{t-1} \mid x_t, c) = \mathcal{N}(x_{t-1} \mid \mu_\theta(x_t, c), \sigma_t^2\mathbf{I}). \tag{7}$$

To guide the diffusion towards the conditional distribution, CFG modifies the noise prediction as:

$$\hat{\epsilon}_\theta(x_t, c) = \epsilon_\theta(x_t, \emptyset) + s \cdot (\epsilon_\theta(x_t, c) - \epsilon_\theta(x_t, \emptyset)) \propto \epsilon_\theta(x_t, \emptyset) + s \cdot \nabla_x \log p(c \mid x_t). \tag{8}$$

During training, conditioning is randomly dropped to learn both conditional and unconditional noise predictions.

Table 9: **Comparison on nuScenes: image quality, spatial consistency, and conditions**

| Method | FID↓ | MVSC↑ | Object Layouts | Camera Poses | Other Conditions |
|---|---|---|---|---|---|
| MagicDrive | 16.20 | 0.8310 | Fourier embedding(1D) | Fourier embedding | Text, map. |
| Panacea | 16.96 | 0.9189 | Perspective projection (2D) | Pseudo-color image | Text, map, depth. |
| **Ours** | 20.70 | 0.9310 | Binary occupancy (3D) | Extrinsic matrix | None |

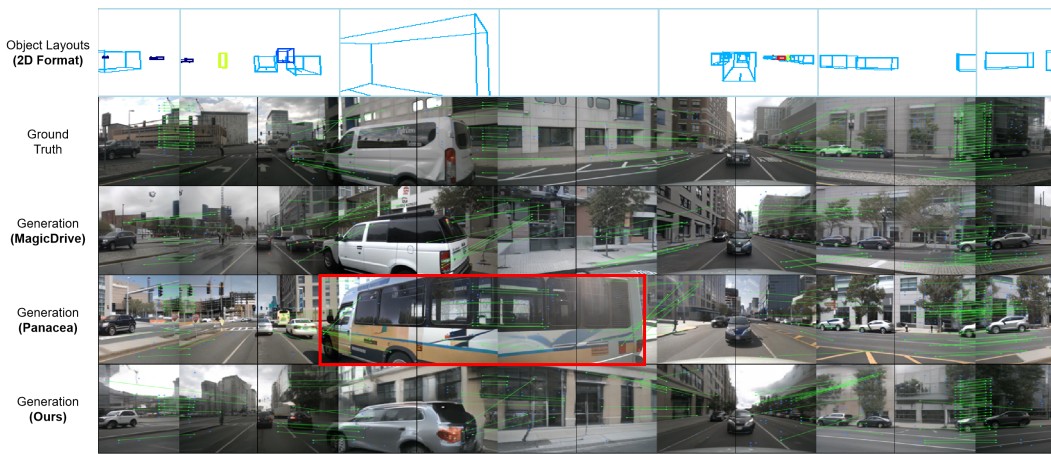

Figure 6: **Multi-View Spatial Consistency (MVSC) on nuScenes.** The comparison is based on images generated by different methods. Row 1 shows the projections of 3D object layouts onto the image plane. Row 2 presents the corresponding validation images. Rows 3–5 display the results generated by MagicDrive, Panacea, and our method, respectively. To better visualize spatial consistency across adjacent views, each row of images is shifted to the right by half an image width. Vertical black lines mark the centerlines of each camera view. Red boxes indicate regions where the generated vehicles are significantly misaligned with the ground-truth layouts.

## B  EVALUATION WITH MULTI-VIEW SPATIAL CONSISTENCY

Evaluating images with pre-trained models is a common practice, with metrics such as Inception Score (IS), Fréchet Inception Distance (FID), and Learned Perceptual Image Patch Similarity (LPIPS) widely used. To assess spatial consistency in multi-view generation, a matching-based metric is introduced. Following prior works such as BEVGen and DriveWM, a pre-trained LoFTR model is employed to perform keypoint matching between adjacent views. Given that the overlapping regions between adjacent views typically cover no more than half of the image centered horizontally, each image is divided vertically into left and right halves. For each adjacent camera pair, keypoint matching is performed between the two bordering half-images, as shown in Fig. 6. The proposed Multi-View Spatial Confidence (MVSC) is then defined as the ratio of this average confidence from reconstructed or generated images to that from real images, serving as an indicator of spatial consistency across views.

Based on the same MVSC metric, Table 9 compares MagicDrive, Panacea, and our method. While our approach yields a higher FID on nuScenes than prior methods, it achieves the best spatial consistency. BEV-VAE adopts a more direct and physically grounded representation of object layouts. MagicDrive encodes 3D boxes with Fourier embeddings and MLPs, fusing them with image features via cross-attention. Panacea projects 3D boxes into the image plane and enforces pixel-level alignment using ControlNet. In contrast, our method represents object layouts as binary occupancy maps in the BEV space, which are inherently aligned with the 3D BEV representation without requiring additional projection or alignment. Camera poses are also handled in a physically consistent way: by rotating the extrinsic matrix applied to the BEV representation, novel views can be rendered directly. This principled 3D-to-2D mapping preserves spatial relationships across views, leading to inherently consistent multi-view generation.

Table 10: FID under different camera rotations on nuScenes.

| Rotation (deg) | -150 | -120 | -90 | -60 | -30 | 0 | 30 | 60 | 90 | 120 | 150 | 180 |
|---|---|---|---|---|---|---|---|---|---|---|---|---|
| FID ↓ | 11.51 | 12.63 | 14.02 | 9.08 | 11.09 | 4.74 | 10.77 | 8.52 | 13.21 | 12.43 | 11.26 | 11.78 |

Table 11: FID under camera translation along the longitudinal direction.

| Translation $x$ (m) | -4 | -2 | -1 | 0 | 1 | 2 | 4 |
|---|---|---|---|---|---|---|---|
| FID ↓ | 9.10 | 5.24 | 4.58 | 4.74 | 4.45 | 5.18 | 9.25 |

## C  FEW-SHOT ADAPTATION FOR MULTI-VIEW RECONSTRUCTION UNDER NEW CAMERA CONFIGURATIONS

We validate this property on WS101, as illustrated in Fig. 7. Specifically, we adapt the pretrained BEV-VAE model to the unseen camera configuration of WS101 using only a small number of training samples, without modifying the model architecture. The results demonstrate that our model can quickly align to new camera intrinsics and extrinsics while preserving strong multi-view consistency and reconstruction quality. These findings highlight the effectiveness of the learned BEV prior in enabling efficient adaptation to novel camera setups.

## D  GENERATION WITH PRECISE 3D OBJECT CONTROL

To demonstrate that the BEV latent space supports precise control based on structured 3D object layouts, we generate multi-view images by selectively removing different vehicles from the same scene. As shown in Fig. 8 and 9, Row 1 presents real images from the validation set, and Row 2 shows the reconstructed images. Row 3 displays images generated from the corresponding 3D bounding boxes. Rows 4–8 further illustrate controllable generation by selectively removing specific vehicles from the input layouts, with the removed objects indicated by numerical labels.

## E  QUANTITATIVE ANALYSIS OF NOVEL VIEW SYNTHESIS

Since camera poses in autonomous driving are fixed, novel-view images do not have ground-truth supervision, making pixel-wise metrics such as PSNR and SSIM inapplicable. We therefore evaluate the perceptual quality of synthesized views using Fréchet Inception Distance (FID), which is widely adopted for generative novel view synthesis without paired supervision.

We conduct a comprehensive evaluation on the nuScenes validation set by rotating the camera in 30° increments over the full 360° range, and translating the camera by 1 m, 2 m, and 4 m along both the $x$ (longitudinal) and $y$ (lateral) directions. As summarized in Table 10, BEV-VAE maintains stable perceptual quality under large camera rotations, while Tables 11 and 12 report the results under longitudinal and lateral translations, respectively, demonstrating strong robustness to spatial perturbations. Notably, the 1 m translated views achieve even lower FID than the original viewpoint, which we attribute to the richer viewpoint diversity induced by our PAS multi-dataset joint training strategy. In addition, we present novel-view synthesis results under translated camera extrinsics in Fig. 10.

## F  MODULE-WISE INFERENCE LATENCY OF BEV-VAE

We evaluate the impact of enabling Flash Attention on the multi-view reconstruction and novel view synthesis speed of BEV-VAE. Table 13 reports the end-to-end inference throughput and peak GPU memory consumption. The results show only marginal improvements in FPS, indicating that deformable attention, rather than standard self-attention, is the primary computational bottleneck in our framework.

To further understand this behavior, we analyze the parameter distribution and per-module latency of BEV-VAE. We decompose the model into its core components, including the image encoder ($\mathbf{E}_{img}$),

Table 12: FID under camera translation along the lateral direction.

| Translation $y$ (m) | -4 | -2 | -1 | 0 | 1 | 2 | 4 |
|---|---|---|---|---|---|---|---|
| FID $\downarrow$ | 9.13 | 5.02 | 4.45 | 4.74 | 4.56 | 5.22 | 8.91 |

Table 13: Effect of Flash Attention on BEV-VAE inference efficiency.

| Model | Flash Attention | FPS | Peak GPU Memory |
|---|---|---|---|
| BEV-VAE | ✗ | 1.94 | 4188.65 MB |
| BEV-VAE | ✓ | 2.00 | 4188.50 MB |

scene encoder/decoder based on deformable attention ($\mathbf{E}_{scn}$, $\mathbf{G}_{scn}$), state encoder/decoder ($\mathbf{E}_{stt}$, $\mathbf{G}_{stt}$), and the reparameterization module. Table 14 reports the number of parameters and the average per-module latency, with and without Flash Attention.

Although the deformable-attention-based scene encoder and decoder ($\mathbf{E}_{scn}$, $\mathbf{G}_{scn}$) contain only a small portion of the total model parameters, they dominate the overall computation time, accounting for the majority of the end-to-end latency. Flash Attention mainly accelerates the standard self-attention layers in the image and state modules, but has little effect on the deformable attention components. Consequently, the overall speedup remains limited. These results indicate that future efficiency improvements should focus on optimizing deformable attention kernels and memory access patterns, rather than solely relying on more efficient self-attention implementations.

## G  EXTENSION TO 512x512 IMAGE RESOLUTION

We adopt a 256×256 input resolution to be consistent with prior VAEs(e.g., SD-VAE, ViT-VQGAN) that employ 8× spatial downsampling. Higher input resolutions are particularly important for autonomous driving, as they enable finer-grained spatial and geometric modeling. BEV-VAE follows the ViT-VQGAN design: the encoder maps a 256×256 image into a 32×32 latent grid using a single convolution layer (patch size = 8), while the decoder reconstructs the image via a single transposed convolution. When scaling to 512×512, the encoder patch size increases from 8 to 16. However, using a single 16× upsampling layer in the decoder is suboptimal. To address this limitation, we replace the decoder head with a lightweight U-Net–style multi-stage upsampling module, denoted as **BEV-VAE***. In addition, we evaluate higher-resolution BEV settings (160×160). Quantitative results are summarized in Table 15.

**Resolution scaling analysis.**  Increasing the input resolution to 512×512 consistently degrades performance. PSNR decreases only slightly, while SSIM drops more noticeably, indicating that reconstructions remain numerically close but lose structural fidelity. MVSC degrades substantially, as higher-resolution images introduce richer view-specific details that make cross-view consistency harder to enforce. Meanwhile, rFID increases sharply, suggesting that high-resolution spatial modeling is significantly more challenging than a naive 2× resolution scaling.

**Loss analysis.**  We analyze the validation losses under different settings. At 512×512, the discriminator loss drops sharply, revealing an adversarial imbalance toward the discriminator. Stronger generators (e.g., higher-resolution 160×160 BEV) are required to restore training balance. Notably, $\mathcal{L}_2$ remains nearly unchanged, which is consistent with the modest PSNR degradation, whereas $\mathcal{L}_{\text{perceptual}}$ increases substantially, aligning with the observed drop in SSIM and increase in rFID.

**Higher BEV resolution.**  Increasing the BEV resolution from 128×128 to 160×160 improves MVSC, indicating stronger multi-view feature fusion and better spatial consistency. However, rFID improves more slowly, likely due to increased model capacity and optimization difficulty introduced by higher-resolution BEV representations.

Table 14: Parameter distribution and per-module latency analysis.

| Module | $\mathbf{E}_{img}$ | $\mathbf{E}_{scn}$ | $\mathbf{E}_{stt}$ | Reparam. | $\mathbf{G}_{stt}$ | $\mathbf{G}_{scn}$ | $\mathbf{G}_{img}$ |
|---|---|---|---|---|---|---|---|
| Params (M) | 85.1 | 5.2 | 42.5 | 0.038 | 42.5 | 2.5 | 85.1 |
| Latency (ms) | 119.10 | 105.50 | 10.31 | 0.36 | 9.30 | 161.86 | 110.06 |
| Latency (ms) w/ FlashAttn | 110.42 | 105.67 | 10.78 | 0.41 | 9.81 | 160.72 | 102.16 |

Table 15: Quantitative comparison under different image and BEV resolutions.

| Model | Training | Image | BEV | PSNR ↑ | SSIM ↑ | MVSC ↑ | rFID ↓ | $\mathcal{L}_{KL}$ | $\mathcal{L}_2$ | $\mathcal{L}_{perceptual}$ | $\mathcal{L}_A$ | $\mathcal{L}_D$ |
|---|---|---|---|---|---|---|---|---|---|---|---|---|
| BEV-VAE | PAS | 256×256 | 128×128 | 28.88 | 0.8028 | 0.9756 | 4.74 | 2.82e4 | 0.031 | 0.15 | 0.942 | 0.331 |
| BEV-VAE | nuScenes | 256×256 | 128×128 | 26.13 | 0.7231 | 0.9250 | 6.66 | 2.40e4 | 0.057 | 0.216 | 0.361 | 0.879 |
| BEV-VAE* | nuScenes | 512×512 | 128×128 | 25.71 | 0.6727 | 0.7729 | 20.54 | 1.95e4 | 0.063 | 0.33 | 3.713 | 0.017 |
| BEV-VAE* | nuScenes | 512×512 | 160×160 | 25.73 | 0.6733 | 0.7823 | 20.99 | 1.93e4 | 0.063 | 0.33 | 4.168 | 0.030 |

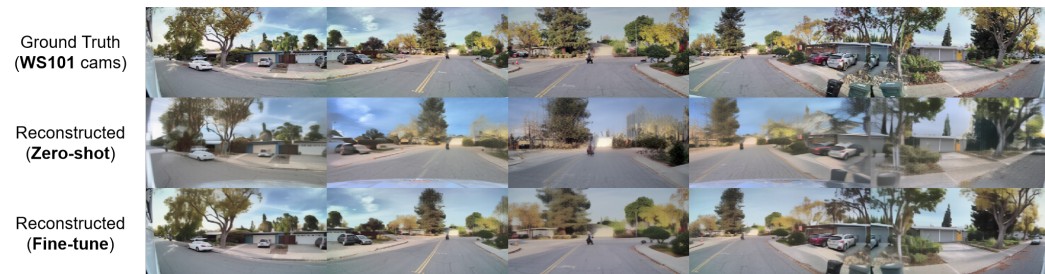

Figure 7: **Few-shot adaptation for multi-view reconstruction on WS101.** Row 1 shows real images from the WS101 validation set. Rows 2 and 3 show zero-shot and fine-tuned reconstructions, respectively, with object shapes preserved in the zero-shot results and further sharpened after fine-tuning.

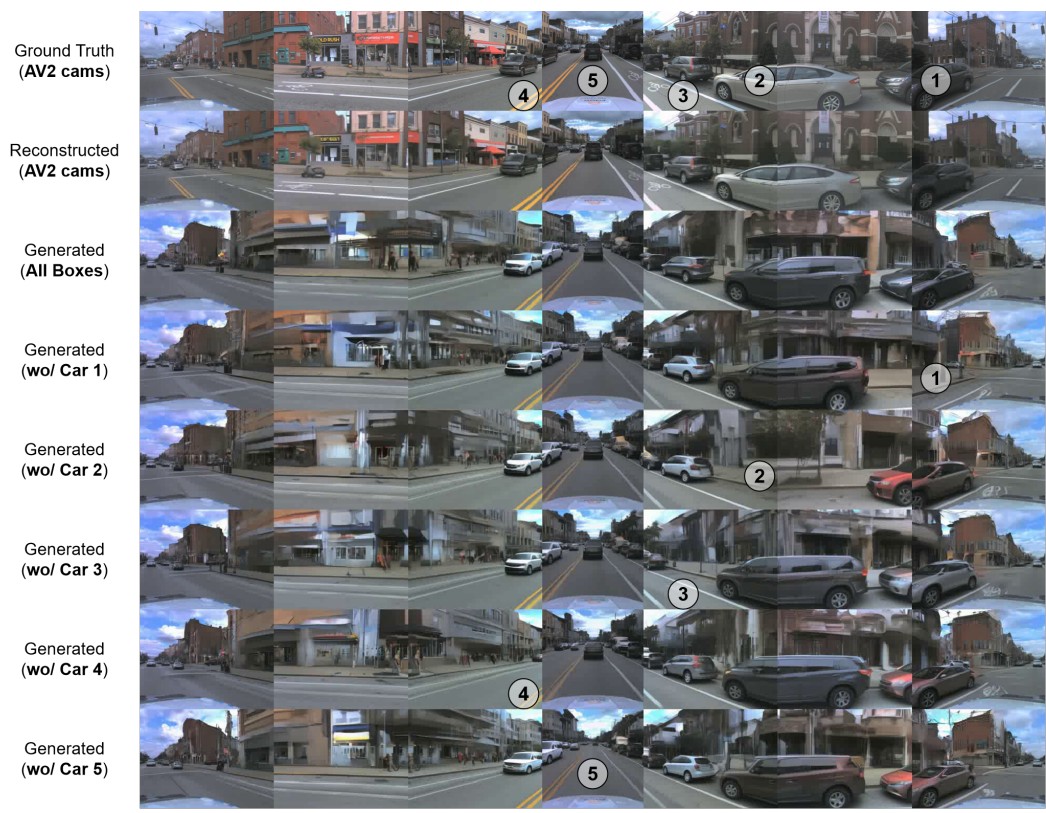

Figure 8: **Multi-view image generation on AV2 with 3D object layout editing.**

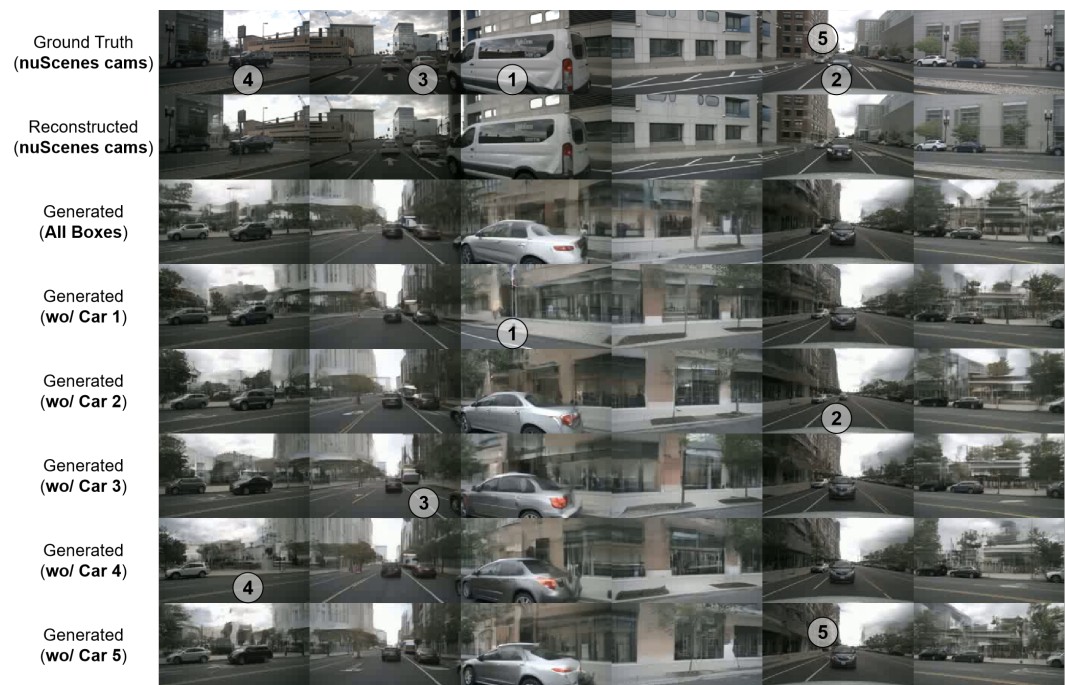

Figure 9: **Multi-view image generation on nuScenes with 3D object layout editing.**

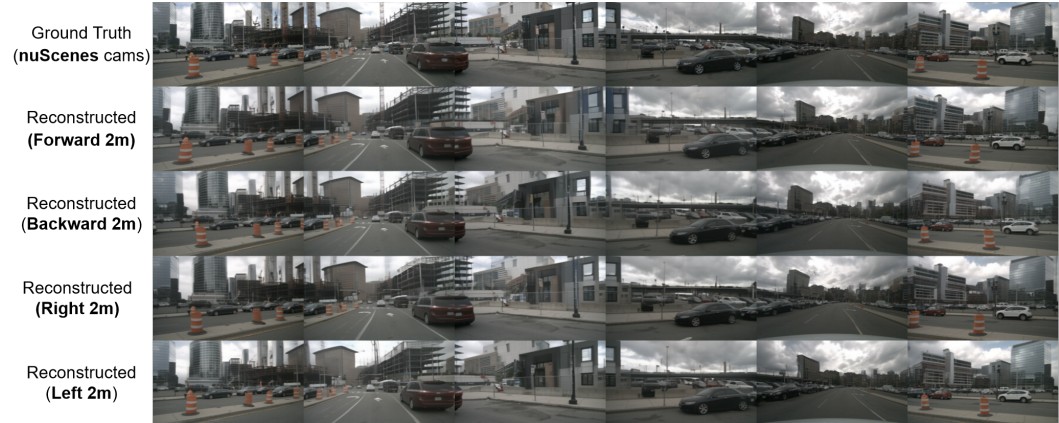

Figure 10: **Novel view synthesis via camera pose modifications on nuScenes.** Row 1 shows real images from the nuScenes validation set. Rows 2–5 show reconstructions with all cameras translated 2 m forward, backward, rightward, and leftward, respectively.

