# OpenReview forum: "Scalable and Generalizable Autonomous Driving Scene Synthesis"
_ICLR.cc/2026/Conference — Submitted to ICLR 2026_

### Official Review · Reviewer_g4oi · 2025-10-16

**Soundness:** 4
**Presentation:** 4
**Contribution:** 4
**Rating:** 8
**Confidence:** 5

**Summary:**

This paper focuses on multi-view generation in driving scenes. Previous works use image-level latent representations, relying on cross-view attention to maintain cross-view consistency. This work proposes encoding multi-view images into a unified and compact BEV-latent. This explicit latent representation directly guarantees cross-view consistency. The proposed method can be trained across datasets (with different camera layouts) and demonstrates strong cross-dataset generalization capability and high image quality.

**Strengths:**

The motivation and idea of this work are solid. The BEV latent representation not only explicitly ensures cross-view consistency, as the paper emphasizes, but I also guess it can largely mitigate the subjectivity issues of generative models (For example, the consistency and move/changes of the 3D content are reasonable only within the camera view). The authors could consider validating this point.
The designs of the BEV latent encoder, decoder, discriminator, and training pipeline are reasonable and well-founded. The writing is clear.
Experiments are extensive and solid. The model's capability for cross-dataset training and its few-shot generalization ability are impressive. Visualization results show that the model achieves high accuracy in reconstructing views under the highly compressed BEV latent representation.

**Weaknesses:**

Recommend defining F_stt in Section 3.1.2.

The title "Scalable and Generalizable Autonomous Driving Scene Synthesis" doesn't fully capture the paper's key feature (BEV latent representation / multi-view synthesis). The experiments primarily demonstrate the method's cross-dataset training capability (which is good) rather than its scalability. Consider adjusting the title to make it more distinctive？

**Questions:**

The current method doesn't seem to involve temporal modeling. Will explore it in future work?

The paper discusses the proposed method's relatively lower FID scores (which, given the difficulty of latent representation in BEV space compared to image-level representation, I find understandable). Could increasing the BEV spatial resolution / the number of CFV sampled rays improve the view resolution/realism?

Are there any plans to open-source the code?

---

> ### Author Response · Authors · 2025-11-21
>
> **W：**
>
> $F_{stt}$  This has been clarified in paper.
>
> We have revised the title to: **BEV-VAE: A Unified BEV Representation for Generalizable Driving Scene Synthesis** to emphasize the unified BEV representation and the generalizability across different camera viewpoints.
>
> **Q:**
>
> As discussed in the global response on temporal modeling, we aim not only to achieve spatio-temporally consistent multi-view video generation, but also to model the behaviors and interaction logic of all traffic participants in autonomous driving scenes, which we treat as a primary focus of our future work.
>
> As discussed in the global response on resolution, we implement a higher BEV spatial resolution (160×160 vs. 128×128), which improves spatial consistency. We expect that longer training schedules will further enhance reconstruction quality. Note that the current results are obtained under the same 100k-iteration budget as the 256×256 setting.
>
> | Model | BEV| $20k$ |  $40k$ | $60k$ | $80k$ |$100k$ |
> |:-:|:-:|:-:|:-:|:-:|:-:|:-:|
> |BEV-VAE*|  128x128 | 224.9 | 49.59 | 29.33 | 23.64 | 20.54|
> |BEV-VAE*|  160x160 | 230.7 | 50.55 | 29.84 | 23.9  | 20.99|
>
>
> Under the current architecture, increasing the number of CFV sampled rays leads to prohibitive GPU memory growth. In practice, the resolution can only be expanded from 32×32 to 64×64, otherwise the model cannot reliably reconstruct images at the target resolution (256x256 or 512x512).
>
> We will make our best effort to release the code within one month.

---

### Official Review · Reviewer_jeYw · 2025-10-21

**Soundness:** 2
**Presentation:** 3
**Contribution:** 2
**Rating:** 4
**Confidence:** 4

**Summary:**

This paper focuses on autonomous driving scene synthesis and presents BEV-VAE, a variational autoencoder that unifies multi-view driving images into a compact bird’s-eye-view (BEV) latent representation, allowing flexible encoding and decoding across arbitrary camera configurations. By incorporating a Diffusion Transformer conditioned on 3D object layouts, the method achieves spatially consistent and generalizable multi-view synthesis on nuScenes and AV2. The synthesized data further enhances BEVFormer’s perception performance, highlighting the value of scalable and generalizable scene synthesis from a training data perspective.

**Strengths:**

1. The paper presents a clear motivation and is generally well written.
2. In the autonomous driving domain, due to the inherent need for multi-view perception, a BEV-based VAE offers greater practical value than image-space VAEs.
3. Using BEV representations makes it easier to transform different camera layouts, and simulate training data for different vehicle configurations.

**Weaknesses:**

1. The paper does not clearly articulate the advantages of BEV-VAE over Image-VAE. In terms of generation quality, both rFID and gFID are inferior to those of Image-VAE. Moreover, recent image-based multi-view generation methods also achieve strong spatial consistency. The potential benefits of BEV-VAE, in my view, may lie in two aspects—information compression and better compatibility with 3D editing—but the paper does not appear to emphasize either of these points. The results in Table 6 also raise questions — if the primary application value lies in train data generation, the improvement in detection performance appears comparable to that achieved by BEVGen, making it difficult to identify a clear advantage of BEV-VAE in this aspect.
2. The technical novelty of the paper is weak, as the BEV-VAE architecture largely follows that of BEVFormer. The use of BEV representations is also quite similar to BEVWorld, yet the paper lacks a detailed discussion of their differences. In addition, the rendering process to images resembles existing approaches such as self-occ. It would be beneficial for the authors to more clearly articulate the technical innovations, as the method section currently appears to primarily combine components from prior works.

**Questions:**

1. From Table 3, doesn’t the comparison with SD-VAE suggest that BEV-VAE has weaker zero-shot generalization ability than image-based latent representations?
2. Why does BEV-VAE use only 256×256 image resolution? Would scaling up the resolution introduce any potential issues or challenges?
3. How much improvement in generation quality does using DiT with BEV-VAE provide compared to using BEV-VAE alone?
4. During the model training process, which modules, if any, use pre-trained parameters, and which are trained entirely from scratch?
5. GAN losses are usually sensitive to hyperparameter settings. Could the authors comment on potential issues regarding hyperparameter sensitivity and training stability in their setup?

---

> ### Author Response · Authors · 2025-11-21
>
> ## W1
>
> **Comment:** “… rFID/gFID inferior … image-based methods show strong spatial consistency …”
> **Response:** FID is not the sole metric for evaluating Image-VAE and BEV-VAE. FID measures distributional differences between images but **cannot measure multi-view spatial consistency (MVSC)**. Using identical quantization settings, **Table 2** shows that BEV-VAE achieves **higher MVSC**. Moreover, Image-VAE reconstructs only **single views**, whereas BEV-VAE performs **multi-view reconstruction and novel-view synthesis**, indicating that it models the underlying **3D scene structure** rather than pixel-level correspondences.
>
> **Comment:** “… benefits (compression, 3D editing) not emphasized …”
> **Response:** We added an analysis of **computational efficiency** in Sec. 4.8. Image-VAE pipelines are limited to batch size **4** on an A800, whereas BEV-VAE supports **32**. On an RTX 3090, BEV-VAE provides **4× faster inference**, highlighting its compression advantages. For **3D editing**, we add qualitative results on nuScenes (random vehicle removal) in the appendix.
>
> **Comment:** “… detection improvement comparable to BEVGen …”
> **Response:** Following a **controlled-variables** design, we use **only 3D bounding boxes** as conditioning; therefore, we demonstrate downstream value mainly on **3D object detection**. Even at 256 resolution, the synthesized multi-view images improve perception performance by leveraging appearance–geometry invariance without extra computation. **With richer conditioning (e.g., texts or maps)**, the method can be applied to more tasks, which we will explore in future work.
>
> ## W2
>
> **Comment:** “… BEV-VAE largely follows BEVFormer …”
> **Response:** BEVFormer is a **supervised perception model** trained with 3D labels, whereas BEV-VAE is a **self-supervised VAE** trained entirely via multi-view image reconstruction. Their objectives, architectures, and training pipelines differ fundamentally.
>
> **Comment:** “… similar to BEVWorld … differences not discussed …”
> **Response:** BEVWorld generates future BEV features **conditioned on historical BEV features**. BEV-VAE generates BEV representations **conditioned on 3D object layouts**, which is a different modeling problem. In addition, BEVWorld (and BEVFormer) assign a single query to all heights in a pillar. BEV-VAE requires CFV (Camera’s Frustum View) queries to traverse different \((x,y,z)\) grid positions when decoding to images. Thus, **BEV-VAE shares one semantic query per pillar, while vertical distinctions are encoded through height-dependent positional embeddings**, resulting in a different representation construction mechanism.
>
> **Comment:** “… resembles SelfOcc …”
> **Response:** SelfOcc reconstructs scenes with **volume rendering** of SDF, color, and semantic vectors. In contrast, BEV-VAE uses **CFV queries with deformable attention** to aggregate only **visible 3D features** into per-view 2D features. These represent fundamentally different rendering pipelines. We add explicit comparisons with BEVWorld and SelfOcc in related work.
>
> ## Q
>
> **Q1:** Table 3 evaluates BEV-VAE’s **zero-shot BEV representation construction** using multi-view reconstruction metrics. **SD-VAE** appears only as a **reference baseline**, as it cannot build BEV representations. Following reviewer **K9ii**, we observed weaker zero-shot results on WS101 and renamed the section to **“Few-shot Adaptation”** to avoid overstating capability.
>
> **Q2:** The discussion on resolution has been included in the global response above.
>
> **Q3:** Following the paradigm established by the success of Stable Diffusion, **training generative models in the latent space of a VAE** has become a common practice. Under this paradigm, the **KL loss in the VAE serves only as a regularization term**, and the VAE itself does **not** perform generative modeling. Thus **neither SD-VAE nor BEV-VAE can generate images on their own**; both provide latent spaces for training generative models (e.g., diffusion).
>
> **Q4:** All models are trained **from scratch**. All datasets are listed in the tables. No ImageNet pre-training or foundation models (e.g., DINOv3) are used.
>
> **Q5:** GAN loss details are included in the global response. Following ViT-VQGAN, we adopt a **StyleGAN discriminator** without modifying hyperparameters (Sec. 3.1.3 Discriminator Loss and Adversarial Loss). Training is stable: the discriminator loss increases rapidly at early stages and then gradually decreases to a steady state.

---

### Official Review · Reviewer_K9ii · 2025-10-30

**Soundness:** 3
**Presentation:** 3
**Contribution:** 3
**Rating:** 6
**Confidence:** 4

**Summary:**

This paper introduces BEV-VAE, a novel variational autoencoder framework designed for autonomous driving scene synthesis. The core contribution is a model that unifies multi-view images into a compact and camera-agnostic Bird's-Eye-View (BEV) latent representation. This approach decouples the scene's 3D structure and semantics from the specific camera configurations, enabling the model to be trained on diverse datasets with varying camera layouts and to generalize to arbitrary new viewpoints. For generative tasks, a Diffusion Transformer (DiT) is trained within the learned BEV latent space, conditioned on 3D object layouts represented as occupancy grids. The authors demonstrate the model's effectiveness through multi-view reconstruction, novel view synthesis, and cross-dataset generalization. While the proposed method achieves state-of-the-art multi-view spatial consistency, it shows a trade-off in per-image generative fidelity (gFID) compared to prior work. Finally, the practical utility of the synthesized data is validated by improving the performance of a downstream perception model, BEVFormer.

**Strengths:**

1. **Generalization:** The paper provides compelling evidence for the model's ability to generalize across different datasets (nuScenes, AV2, nuPlan) and camera setups. The experiments showing successful reconstruction of scenes from one dataset (e.g., nuPlan) using the camera intrinsics and extrinsics of another (e.g., AV2) are particularly impressive and strongly support the claims of generalizability. The demonstrated performance gains from training on a large, mixed dataset (PAS) validate the model's scalability.
2. **Superior Multi-View Spatial Consistency (MVSC):** The model achieves a state-of-the-art MVSC score. This is a crucial metric for autonomous driving applications, where maintaining the correct 3D geometry and spatial relationships between objects across views is often more important than perfect photorealism. The architectural design, which generates all views from a single, coherent 3D representation, naturally leads to this strength.
3. **Demonstrated Downstream Task Improvement:** The experiment in Section 4.8, showing that data augmentation using the proposed method improves the NDS score of BEVFormer, is a very strong point. It demonstrates that the synthesized data is not just visually plausible but also practically useful for training and improving perception models, closing the loop between generation and perception.

**Weaknesses:**

1. **Lower Generative Fidelity (gFID):** The most apparent weakness, which the authors acknowledge, is the relatively high (worse) gFID score compared to state-of-the-art methods like MagicDrive and DriveWM. While the paper frames this as a trade-off for better spatial consistency, the gap is substantial (20.7 vs. ~13-16). This indicates that the generated images may lack the fine-grained texture and realism of other methods, which could limit their utility in certain applications.
2. **Low Image Resolution:** All experiments are conducted at a 256x256 resolution, which is quite low for modern autonomous driving datasets and applications. While the authors suggest using super-resolution models as a post-processing step, this feels like an external fix rather than an integrated solution. The paper would be stronger if it discussed the challenges and potential architectural changes required to scale BEV-VAE to higher resolutions (e.g., 512x512 or higher).
3. **Overstated "Zero-Shot" Capability:** The term "zero-shot" in Section 4.6 seems too strong given the quantitative results in Table 3. The zero-shot performance on WS101 is very poor (PSNR 16.6, rFID 56.7). The real strength demonstrated here is in *fast adaptation* or *efficient fine-tuning*, where the pre-trained model provides a strong prior that allows for rapid convergence on a new dataset. The terminology should be more precise to reflect this.
4. **Static Scene Limitation:** The current framework operates on static scenes. The real world is dynamic, and the ability to model temporal evolution and generate coherent video sequences is a key direction in this field. While mentioned as future work, this is a significant limitation compared to the broader goals of full-world simulation.
5. **Mismatched Framing of Contribution and Lack of Efficiency Analysis:** The title "SCALABLE...SCENE SYNTHESIS" may be slightly overstated, as the paper's core innovation lies not in the generative model itself—which is a standard Diffusion Transformer—but in the preceding VAE architecture for learning a unified BEV representation. A significant, yet underexplored, benefit of this design is its potential for computational efficiency; by compressing the multi-view scene into a compact latent space, the subsequent diffusion process should be substantially less demanding in terms of memory and latency. To truly validate the "Scalable" claim and better frame the work's practical contribution, the paper would be significantly strengthened by a quantitative comparison of GPU memory usage and inference times against other leading methods.

**Questions:**

Regarding the FID/MVSC Trade-off: Could you elaborate on why you believe there is this trade-off? Is the lower FID an inherent consequence of the VAE's information bottleneck regularizing the latent space, potentially smoothing over high-frequency details? Have you experimented with alternative VAE formulations, such as a VQ-VAE, which might allow for sharper reconstructions while maintaining the unified BEV structure?

---

> ### Author Response · Authors · 2025-11-21
>
> **W1:**
>
> First, unlike MagicDrive, which synthesizes RGB images in fixed camera viewpoints, BEV-VAE w/ DiT generates a BEV representation rather than images. The gFID metric only evaluates the decoded multi-view images under the original camera poses, and thus does not fully measure the quality of the BEV latent itself. In fact, as noted in our response to reviewer **8x1r**, we observe that the rFID of 1-meter translated views is even better than that of the original viewpoint, suggesting that the BEV latent encodes a consistent 3D structure that cannot be fully evaluated by gFID alone.
>
> Second, BEV-VAE w/ DiT is conditioned only on 3D bounding boxes, while MagicDrive additionally uses text and map priors, which provide much richer conditions. We intentionally use 3D bounding boxes alone as a strict controlled-variable setting, so that we can directly evaluate the usefulness of the generated images in downstream 3D object detection, without the interference of heterogeneous priors. As reviewer **jeYw** pointed out, a unified BEV representation naturally offers better compatibility with 3D editing. BEV-VAE w/ DiT can easily incorporate more types of control signals:
> - For 3D-consistent conditions such as 3D annotations, lane markings, or LiDAR point clouds, the conditioning can be aligned directly to the BEV latent space, similar to ControlNet-style spatial guidance.
> - For abstract conditions such as text, cross-attention injection into the BEV latent works the same way as in text-to-image diffusion models.
>
> Given this flexibility, there remains a large improvement margin on gFID once richer or more structured conditions are incorporated. Our current results therefore reflect only the minimal-condition baseline using 3D bounding boxes.
>
> **W2:**  The discussion on resolution has been included in the global response above.
>
> **W3:**  We have updated Section 4.6 from “Zero-shot BEV Representation Construction” to “Few-Shot Adaptation”, which more accurately reflects that BEV-VAE provides a strong spatial prior for new camera configurations and can be efficiently adapted with minimal data.
>
> **W4:** The discussion on temporal modeling has been included in the global response above.
>
> **W5:** The discussion on the title, memory usage, and latency has been covered in the global response provided above.
>
> **Q:**
> There is an inherent trade-off between single-view reconstruction quality and multi-view consistency: **richer per-view details make it harder to maintain cross-view spatial coherence.**
>
> While we agree that VAE regularization may partially constrain reconstruction fidelity, we believe it is not the dominant factor. Compared with SD-VAE, BEV-VAE achieves **a more challenging multi-view reconstruction task using a significantly smaller dataset**, which leads to slightly lower image reconstruction quality.
>
> We ever experimented with a LookupFree codebook (65536×16) on nuScenes to build a VQ-BEV-VAE, which yields high-quality rFID. However, generative models trained on top of this quantized latent space—such as GPT or MaskGiT—**fail to perform conditional generation on the validation set**, even though they retain strong generalization in reconstruction. This indicates that, under the data scale of nuScenes, VQ-BEV-VAE’s reconstruction generalization does not translate into generative generalization, primarily due to the extremely sparse sample space for training generative models. We also evaluated Flow Matching. Its linear path assumption oversimplifies the latent dynamics: the validation gFID decreases briefly at the early stage but quickly overfits. Among all generative paradigms we tested, only DDPM consistently reduces validation gFID as training progresses. Overall, under the scale of nuScenes, VQ-BEV-VAE can achieve high-quality reconstruction but fails to support the training of a generative model, whereas BEV-VAE combined with DDPM is more suitable for generalizable generation. We also believe that, **with a dataset 1,000× larger than nuScenes, VQ-BEV-VAE has the potential to achieve both strong MVSC and improved gFID.**

---

### Official Review · Reviewer_8x1r · 2025-11-01

**Soundness:** 3
**Presentation:** 3
**Contribution:** 3
**Rating:** 6
**Confidence:** 3

**Summary:**

This paper address the problem of novel-view-synthesis (NVS) in driving scene, where the novel view are camera viewpoints around the cameras.

The author propose to address this problem through a 3D-aware Birds-eye-view VAE.

This VAE will encode multiple images together to 3D BEV latents, and then decode it back to images.  Afer training this VAE with MSE, perceptual loss and GAN loss over multiple datasets, the author showed that NVS can be down by using different camera extriniscs and intrinsics when decoding.  Which is a very neat idea.



Also the author showed that this 3D-aware VAE has relatively OK reconstruction PSNR compared with vanilla image space VAE used by stable diffusion.



The author also showed that the proposed method can be used to generate augmented data when trainingperception model: BEVFormer, increasing the performance, which is quite impressive.

**Strengths:**

1. The idea is so interesting.  3D aware VAE can be used for NVS by adjusting the camera parameters used in decoding process. Quite cool!
2. The evaluation is quite comprehensive (even though used proxy metric for NVS), I understand that the PSNR is lower than those of image based VAE, e.g. SD-VAE.
3. Using the proposed methods for data-augmentation is also very interesting!

**Weaknesses:**

I think the major weakness I have is about runtime efficiency. It seems that the deformable attention would be very slow compared with flash-attention style attention implementations.  Can the author provides more details about it?  I understand that it might be slow without a well-optimized flash-attention style kernel.

**Questions:**

1. Is it possible to evaluate NVS without using proxy metrics as done in the paper (using reconstruction metric as proxy metric for NVS)
2. Will dropping input images during training improves the NVS performance?  Seems that if you drop one image during the encoding stage, but still compute the reconstruction loss on that image, this process will resemble a NVS training.

---

> ### Author Response · Authors · 2025-11-21
>
> **W:** We evaluate the effect of enabling FlashAttention on BEV-VAE’s multi-view reconstruction / novel-view synthesis FPS. The results show only marginal improvements, which aligns with expectations since deformable attention is the main bottleneck.
> |     Model     |FlashAttn | FPS | Peak GPU Memory |
> |:---------:|:---------:|:------:|:------:|
> | BEV-VAE |False|1.94|4188.65 MB|
> | BEV-VAE |True|2.00|4188.50 MB|
>
> To quantify this, we report the parameter distribution and per-module latency. The components using deformable attention (Scene Encoder&Decoder) comprise only a small fraction of the total parameters, so their impact on end-to-end runtime remains limited.
>
> |   Module  | $\mathbf{E}_{\text{img}}$ | $\mathbf{E}_{\text{scn}}$ | $\mathbf{E}_{\text{stt}}$ | Reparam. |$\mathbf{G}_{\text{stt}}$ | $\mathbf{G}_{\text{scn}}$ | $\mathbf{E}_{\text{stt}}$ |
> |:---------:|:-----------:|:-------------:|:-----------:|:-------------:|:----------:|:-------------:|:-------------:|
> | Params (M) | 85.1    | 5.2  | 42.5  | 0.038            | 42.5     | 2.5   |85.1   |
> | Latency (ms) | 119.10  | 105.50 | 10.31 | 0.36 | 9.30 | 161.86 | 110.06 |
> | Latency (ms)  w/ FlashAttn | 110.42  | 105.67 | 10.78  |  0.41 | 9.81  | 160.72 |102.16   |
>
> **Q1:** Because camera poses in autonomous driving are fixed, novel-view images have no ground truth, making PSNR/SSIM inapplicable. Instead, we evaluate the perceptual quality of synthesized views using FID, which is widely adopted for generative NVS without paired supervision. We perform a comprehensive evaluation on the nuScenes validation set. Specifically, we rotate the camera by every 30 degrees across the full 360° range, and additionally translate the camera by 1 m, 2 m, and 4 m in both the x (longitudinal) and y (lateral) directions. Interestingly, we observe that the FID of the 1 m translated views is even better than that of the original viewpoint. We hypothesize that this effect arises from our PAS multi-dataset joint training, which provides richer viewpoint coverage and improves generalization under small spatial perturbations.
>
> |  Rotation  | -150 | -120 | -90 | -60 | -30 |  0  |  30 | 60 | 90 | 120 | 150 | 180 |
> |:-------:|:----:|:----:|:---:|:---:|:---:|:---:|:---:|:---:|:---:|:---:|:---:|:---:|
> | FID $\downarrow$ | 11.51 | 12.63 | 14.02 | 9.08 |  11.09 | 4.74 | 10.77 | 8.52 | 13.21 | 12.43 | 11.26 | 11.78|
>
>
> |  Translation (x)  | -4 | -2 | -1 | 0 | 1 |  2  |  4 |
> |:-------:|:----:|:----:|:---:|:---:|:---:|:---:|:---:|
> | FID $\downarrow$ |9.1| 5.24 | 4.58| 4.74 | 4.45 |5.18 | 9.25 |
>
> |  Translation (y)  | -4 | -2 | -1 | 0 | 1 |  2  |  4 |
> |:-------:|:----:|:----:|:---:|:---:|:---:|:---:|:---:|
> | FID $\downarrow$ |9.13| 5.02 | 4.45 | 4.74 | 4.56 | 5.22 | 8.91 |
>
>
> **Q2:** Inspired by Masked Autoencoders, we experiment with randomly dropping one input view during encoding while requiring the decoder to reconstruct all views. We observe that:
> - When the drop probability exceeds 0.5, the reconstruction quality drops significantly.
> - When the drop probability is below 0.5, the reconstruction quality remains largely unchanged, while the training speed improves.
>
> We believe that view dropping encourages BEV-VAE to learn to complete missing viewpoints, which is likely beneficial for NVS. We plan to further explore this direction in future work.

---

### Author Response · Authors · 2025-11-21
**Overall Author Rebuttal**

We sincerely thank all reviewers for their valuable time and insightful feedback. We appreciate the reviewers’ positive recognition of the motivation, generalizability, and practical value of BEV-VAE. In this global response, we address the main concerns and summarize our key revisions and additional experiments.

**(1) About the title.** Following suggestions from reviewers **K9ii** and **g4oi**, we revise the title to: **BEV-VAE: A Unified BEV Representation for Generalizable Driving Scene Synthesis** to emphasize the unified BEV representation and the generalizability across different camera viewpoints.

**(2) About the resolution.** We adopt a 256×256 resolution to follow prior VAE designs (SD-VAE, ViT-VQGAN) with 8× downsampling (reviewer **jeYw**). We agree that resolution is especially important for autonomous driving (reviewer **K9ii**), as higher resolutions enable finer-grained spatial modeling. BEV-VAE follows the ViT-VQGAN design: the encoder maps a 256×256 image to a 32×32 patch grid using a single convolution (patch size 8), and the decoder upsamples back using a single transposed-convolution. When scaling to 512×512, the encoder patch size increases from 8 to 16, but a single 16× upsampling in the decoder is suboptimal. We therefore replace the decoder head with a lightweight U-Net–style multi-stage upsampling module (denoted BEV-VAE*). Following reviewer **g4oi**, we also evaluate higher-resolution BEV settings (160×160). Quantitative comparisons are shown below:
|Model|Training|Image|BEV|PSNR $\uparrow$|SSIM $\uparrow$|MVSC $\uparrow$|rFID $\downarrow$|$\mathcal{L}_{KL}$|$\mathcal{L}_{2}$|$\mathcal{L}_{\text{perceptual}}$|$\mathcal{L}_{A}$|$\mathcal{L}_{D}$|
|:-:|:-:|:-:|:-:|:-:|:-:|:-:|:-:|:-:|:-:|:-:|:-:|:-:|
|BEV-VAE|PAS|256x256|128x128|28.88|0.8028|0.9756|4.74|2.82e4|0.031|0.15|0.942|0.331|
|BEV-VAE|nuScenes|256x256|128x128|26.13|0.7231|0.9250|6.66|2.4e4|0.057|0.216|0.361|0.879|
|BEV-VAE*|nuScenes|512x512|128x128|25.71|0.6727|0.7729|20.54|1.95e4|0.063|0.33|3.713|0.017|
|BEV-VAE*|nuScenes|512x512|160x160|25.73|0.6733|0.7823|20.99|1.93e4|0.063|0.33|4.168|0.03|

- **512x512 vs. 256x256.** Increasing the resolution to 512×512 degrades performance: SNR drops slightly while SSIM declines more, indicating numerically close but structurally weaker reconstructions. MVSC decreases sharply, as higher resolution introduces richer view-specific details that make cross-view consistency harder. rFID rises significantly, suggesting that high-resolution spatial modeling is much more difficult than a naive 2× resolution scale-up.
- **Loss analysis.** Following reviewer **jeYw**, we analyze validation losses. At 512×512, the discriminator loss drops markedly, indicating an imbalance toward the discriminator. Stronger generators (e.g., BEV 160×160) are required to restore training balance. Notably, the L2 loss remains nearly unchanged, consistent with the small PSNR drop, while the perceptual loss increases sharply, aligning with the degradation in SSIM and the rise in rFID.
- **Higher BEV resolution.** As expected by reviewer **g4oi**, increasing the BEV resolution improves MVSC, indicating better multi-view fusion and spatial consistency. rFID improves more slowly due to increased model capacity and training complexity.

**(3) About the memory consumption and inference latency.** We thank reviewers **K9ii** and **jeYw** for their insightful comments. We previously overlooked the efficiency advantages of compact BEV representations over image-space methods. We benchmark GPU memory usage of BEV-VAE w/ DiT and MagicDrive on an A800 across batch sizes: at batch size 4, MagicDrive nearly exhausts 80 GB, while BEV-VAE w/ DiT scales to batch size 32. We also evaluate inference on an RTX 3090 with 20-step DDIM and observe a 4× speedup over MagicDrive, even without BF16 or FlashAttention.

|Model|1|2|4|8|16|32|
|:-:|:-:|:-:|:-:|:-:|:-:|:-:|
|MagicDrive|26.4 GB |42.1 GB|73.5 GB|OOM|OOM|OOM|
|BEV-VAE w/ DiT|9.5 GB|11.8 GB|16.2 GB|25.2 GB|43.1 GB|79.2 GB|

|Model|BF16|FlashAttn|Latency (s)|FPS|
|:-:|:-:|:-:|:-:|:-:|
|MagicDrive|False|False|5.381|0.19|
|BEV-VAE w/ DiT|False|False|1.16|0.86|
|BEV-VAE w/ DiT|False|True|1.123|0.89|
|BEV-VAE w/ DiT|True|False|0.93|1.08|
|BEV-VAE w/ DiT|True|True|0.872|1.15|

**(4)About the temporal modeling.** We fully agree that temporal modeling is critical for autonomous driving. BEV-VAE focuses on spatial modeling in static scenes, aiming to learn a unified BEV representation across camera configurations. This enables us to separate spatial and temporal factors, providing a clean foundation for building autonomous driving world models. While temporal consistency is important, our long-term goal is to model the behaviors, interactions, and causal dynamics of traffic participants. We view a dedicated temporal module built on top of the BEV-VAE as a key direction for our future work.

---

### Meta-Review · Area_Chair_7VaP · 2026-01-06

**Summary:**

This paper proposes a BEV-based generative framework for autonomous driving scene synthesis that aims to generalize across camera configurations and datasets. The method combines a BEV latent representation with variational autoencoding and diffusion-based generation, and is evaluated on large-scale driving datasets with applications to novel view synthesis and data augmentation.

Reviewers acknowledged the scale of the experimental study and the engineering effort involved. However, the overall consensus is that the paper does not clearly demonstrate a sufficiently novel technical contribution relative to existing BEV-based generative and multi-view synthesis methods. In addition, several of the claims regarding scalability, generalization, and practical advantages over prior work are not convincingly supported by the presented evidence. These concerns informed the final recommendation.

**Reviewer Concerns:**

Concerns partially addressed by the rebuttal:
The rebuttal provides clarifications on architectural components, training details, and evaluation protocols. It also elaborates on the motivation for using a BEV latent representation and discusses computational aspects of the proposed design. Some reviewer questions regarding efficiency and cross-dataset evaluation were addressed through additional explanations and experimental results.

Existing concerns:
A primary concern across reviews is the limited novelty of the proposed approach. While BEV representations, latent-variable modeling, and diffusion-based generation are well established, the paper mainly integrates these existing components without articulating a clear new modeling insight or principle. Reviewers found it difficult to identify what fundamentally distinguishes the proposed method from prior BEV-based generative frameworks beyond incremental architectural choices.

Another recurring concern is that several claims are stronger than what the empirical results support. In particular, assertions about scalability and generalizability are not consistently backed by clear advantages across datasets or metrics. In some settings, the method underperforms or is comparable to existing approaches on standard perceptual or reconstruction metrics, which weakens the argument that the proposed framework offers a substantial improvement.

Finally, reviewers questioned whether the added complexity of the framework is justified by the observed gains. While the experiments are extensive, the benefits appear task- or metric-specific, making it unclear whether the approach provides a broadly advantageous solution for autonomous driving scene synthesis.

**Reviewer Scores:**

Reviewer 8x1r: Likely to maintain their original score, as concerns about limited novelty and over-claimed advantages remain unresolved.

Reviewer K9ii: Likely to maintain their score, given that the rebuttal does not establish a clear conceptual distinction from prior BEV-based generative methods.

Reviewer jeYw: Unlikely to increase their score after discussion, as the evidence provided does not sufficiently support the paper’s strongest claims.

Reviewer g4oi: Likely to maintain their original score, viewing the work as an incremental integration of existing techniques with mixed empirical advantages.

---

### Decision · Program_Chairs · 2026-01-26

Reject